# A Review of Advanced Cutaneous Melanoma Therapies and Their Mechanisms, from Immunotherapies to Lysine Histone Methyl Transferase Inhibitors

**DOI:** 10.3390/cancers15245751

**Published:** 2023-12-08

**Authors:** Renato Santos de Oliveira Filho, Daniel Arcuschin de Oliveira, Melissa Maeda Nisimoto, Luciana Cavalheiro Marti

**Affiliations:** 1Department of Plastic Surgery, Escola Paulista de Medicina–Universidade Federal de São Paulo–EPM-UNIFESP, São Paulo 04023-062, SP, Brazil; 2Department of Plastic Surgery, Universidade Federal de São Paulo–UNIFESP-Skin Cancer and Melanoma Fellow, São Paulo 04023-900, SP, Brazil; daniel.arcuschin@unifesp.br; 3Cutaneous Oncology Service, Hospital Nove de Julho, São Paulo 01409-002, SP, Brazil; melissa.maeda@unifesp.br; 4Experimental Research Department, Hospital Israelita Albert Einstein, São Paulo 05652-900, SP, Brazil

**Keywords:** advanced melanoma, immunotherapy, target therapy, lysine histone methyl transferase inhibitors, UNC0642

## Abstract

**Simple Summary:**

Advanced cutaneous melanoma is the most aggressive type of skin cancer and has variable rates of treatment response. Currently, there are some classes of immunotherapy and target therapies for its treatment. Immunotherapy can inhibit tumor growth and its recurrence by triggering the host’s immune system, whereas targeted therapy inhibits specific molecules or signaling pathways. However, melanoma responses to these treatments are highly heterogeneous, and frequently become resistant. Thus, this review article presents newly available therapies and those in development for advanced cutaneous melanoma therapy.

**Abstract:**

Advanced cutaneous melanoma is considered to be the most aggressive type of skin cancer and has variable rates of treatment response. Currently, there are some classes of immunotherapy and target therapies for its treatment. Immunotherapy can inhibit tumor growth and its recurrence by triggering the host’s immune system, whereas targeted therapy inhibits specific molecules or signaling pathways. However, melanoma responses to these treatments are highly heterogeneous, and patients can develop resistance. Epigenomics (DNA/histone modifications) contribute to cancer initiation and progression. Epigenetic alterations are divided into four levels of gene expression regulation: DNA methylation, histone modification, chromatin remodeling, and non-coding RNA regulation. Deregulation of lysine methyltransferase enzymes is associated with tumor initiation, invasion, development of metastases, changes in the immune microenvironment, and drug resistance. The study of lysine histone methyltransferase (KMT) and nicotinamide N-methyltransferase (NNMT) inhibitors is important for understanding cancer epigenetic mechanisms and biological processes. In addition to immunotherapy and target therapy, the research and development of KMT and NNMT inhibitors is ongoing. Many studies are exploring the therapeutic implications and possible side effects of these compounds, in addition to their adjuvant potential to the approved current therapies. Importantly, as with any drug development, safety, efficacy, and specificity are crucial considerations when developing methyltransferase inhibitors for clinical applications. Thus, this review article presents the recently available therapies and those in development for advanced cutaneous melanoma therapy.

## 1. Introduction

Melanocytic neoplasms have benign lesions, such as melanocytic nevi, and malignant ones, such as melanomas. Melanocytes are neural crest-derived cells that, during development, colonize the skin, eye, and, to a lesser extent, several other tissues throughout the body. Melanocytes at these diverse sites can originate phenotypically diverse types of melanomas, which are considered the most aggressive type of skin cancer due to their ability to spread rapidly through the body [1]. 

The most frequent types of melanomas in Caucasians are observed on sun-exposed skin. These cutaneous melanomas can be categorized by their origins from skin that is chronically exposed to the sun and skin that is not. 

Excessive exposure to the sun’s ultraviolet radiation (modifiable factor) is considered the main exogenous risk factor for cutaneous melanoma development. This includes prolonged exposure to the sun without adequate protection, as well as the use of tanning beds that produce sun-like radiation. Long-term exposure to UV radiation causes a degenerative change in the elastic fibers of the dermis [1]. Thus, this type of melanoma typically originates from the head, neck, and dorsal surfaces of the distal extremities of older individuals (>55 years of age). They possess a high mutational burden and are associated with neurofibromin 1 (NF1), NRAS, and BRAFnonV600E [1].

Instead, for skin not chronically exposed to the sun, the risk factor for melanoma can be endogenous, such as genetic susceptibility which is accountable for 8% to 10% of cases, with cyclin-dependent kinase inhibitor 2A (CDKN2A) as the main high-risk gene for melanoma [2]. Alternatively, random errors in DNA replication (unmodifiable factor) can also contribute as an intrinsic risk factor for disease development [2].

Melanomas in situ concern a proliferation of melanocytes with enlarged nuclei that are increased in an irregular pattern entirely within the epidermis. This pattern is found at the borders of invasive primary melanomas but can also be observed in free-standing lesions without invasive components. These latter lesions are staged as Tis, the earliest clinical stage of melanoma, according to the American Joint Committee on Cancer [3]. The survival rate of completely resected melanoma in situ at this stage is nearly 100% [4,5].

Melanomas in situ on skin damaged by chronic sun exposure (lentigo maligna) generally arise de novo, without any associated nevus that could have served as precursor lesions [6]. The cell basis of melanomas in situ is not known. However, the high mutation burden indicates that it is a superficially localized melanocyte that may belong to the interfollicular epidermis. In addition, lentigo maligna frequently involves the hair follicles, making the bulge region of the hair follicle an alternative site for its cell source. Melanocytes in other annexal structures, such as the eccrine sweat glands, have also been shown to harbor cells of acral melanoma origin [5].

Once melanoma cells leave the epidermis epithelium and enter the subjacent mesenchymal tissue, such as the dermis, the melanoma becomes invasive. In contrast to many epithelial neoplasms, the ability to invade the dermis alone is not a malignant feature by itself, as most nevi are associated with melanocytes in the dermis. For melanomas, the risk of metastatic disease and death correlates with the degree of invasion [5]. Most invasive melanomas develop from melanomas in situ, which can have different features, depending on the type of melanoma associated with them. Melanomas are termed metastatic once their cells have disseminated outside the local site of the primary tumor and invaded other tissues. Like many solid tumors, melanoma metastases generally appear first in the lymph nodes of the draining area of the primary tumor, whereas distant metastases involving visceral sites tend to appear later. 

However, circulating tumor cells are usually found in patients who display only regional metastasis, or no metastases [7,8], demonstrating that once a melanoma is metastatic, it is very difficult to achieve complete surgical resection or remission as seen in situ melanomas, evidencing the need for new strategies to treat these advanced cases of melanoma.

Thus, despite advances made with immunotherapy and targeted therapy, most patients do not respond to or become resistant to these therapies and still die of disease. New alternative treatments are needed for these patients. Thus, in this review, we will assess therapies, their mechanisms, and their combinations for the treatment of advanced cutaneous melanoma, from immunotherapies to methyl transferase inhibitors.

However, early disease detection is essential for the successful treatment of cutaneous melanoma, independent of the treatment type.

## 2. Advanced Melanoma Treatments

### 2.1. Immunotherapy

Immunotherapy can inhibit tumor growth and its recurrence by triggering the host immune system or improving the antitumor response. Thus far, several immunotherapy drugs, involving mainly the blockade of negative immune checkpoints, have been approved by the Food and Drug Administration (FDA) for metastatic melanoma [9]. Currently, immunotherapy has achieved remarkable results in treating advanced cutaneous melanoma. Some patients experience a lasting response, with tumors significantly decreasing in size or even disappearing completely [10,11,12,13]. 

Three classes of immune checkpoint inhibitors are available for clinical use: inhibitors of the Cytotoxic T-Lymphocyte Associated Protein 4 (CTLA4) receptors (ipilimumab), inhibitors of the Programmed Cell Death-1 (PD-1) co-receptor (nivolumab, spartalizumab, pembrolizumab, and toripalimab) or inhibitors of their ligand PD-L1 (atezolizumab), and inhibitors of the Lymphocyte-Activation Gene 3 (LAG3) transmembrane protein (relatlimab) or the combination of anti-PD-1 with anti-LAG3 (opdualag). These proteins are mainly expressed by T cells but are not restricted to these cells. 

CTLA-4 and CD28 are both important stimulatory receptors involved in regulating T-cell activation. They are present in T cells and share the common ligands B7.1 and B7.2 present in antigen-presenting cells (APC). CTLA4 prevents the stimulatory signaling of T cell proliferation provided by the binding of CD28 with the co-stimulatory molecules, B7.1 and B7.2, during the immune response priming phase. The expression and biological function of CTLA-4 are important for the immune response to negative feedback which regulates T-cell responses. Therefore, the elimination of CTLA-4 can result in the breakdown of immune tolerance [14] (Figure 1a).

In contrast, the immune evasion process controlled by the PD-1/PD-L1 axis is primarily attributed to the overexpression of PD-L1 on cancer cells, which binds to PD-1 expressed on antigen-stimulated T cells and inhibits the activity of the PI3K/AKT and RAS/MEK/ERK signaling pathways. Consequently, this inhibition impairs the proliferation, differentiation, and activation of T cells during the effector phase. Blocking either PD-1 or PD-L1 will result in the breakdown of the immune tolerance induced after T-cell stimulation [14] (Figure 1a).

Like PD-1 and CTLA-4, LAG-3 is not expressed on naive T cells, but its expression can be induced on T cells upon antigen stimulation. LAG-3 has been proposed to bind to the major histocompatibility complex II (MHCII) with higher affinity than T cell receptor antigen (TCR). Thus, LAG-3 inhibits T-cell activation by interfering with the association of TCR with MHCII present in APC. In addition, another LAG-3 proposed ligand is fibrinogen-like protein 1 (FGL1), a molecule secreted from hepatocytes in the liver under normal physiological conditions, which some tumor cells can also produce at high levels. FGL1 has been demonstrated to reduce the secretion of IL-2 in cells expressing LAG-3 upon stimulation with the cognate peptide. Blocking LAG-3 results in the breakdown of immune tolerance induced after T-cell stimulation [15] (Figure 1a).

The response to immunotherapy varies from person to person, and not all patients experience the same results. Approximately 60% of patients develop resistance to immunotherapy treatment and their disease returns and progresses [16]. In addition, it is important to highlight that immunotherapy can also be associated with significant side effects because it activates the immune system more intensely. These side effects can alternate from mild symptoms, such as fatigue and nausea, to more serious immune reactions that can affect specific organs. Therefore, proper administration and monitoring are essential during this type of treatment [17,18,19]. 

Thus, finding new combinatory-efficient treatments is essential. In this sense, LAG-3 plays a vital role in immune homeostasis maintenance and tumor immune escape and is widely present in various activated immune cells [20,21]. The combination of a PD-1 and LAG-3 blockade may be a new treatment for drug-resistant patients [22]. There is a report of a Phase 2–3, global, double-blind, randomized trial, that evaluated relatlimab (LAG-3) and nivolumab (PD-1) as a fixed-dose combination, compared with nivolumab alone, administered intravenously every four weeks to patients with previously untreated metastatic or unresectable melanoma. The inhibition of two immune checkpoints, LAG-3 and PD-1, provided a greater benefit in progression-free survival than the inhibition of PD-1 alone in patients with previously untreated metastatic or unresectable melanoma [13].

There are also clinical studies on other types of immunotherapies such as therapeutic mRNA vaccines, which induce the production of tumoral immunogenic proteins by antigen-presenting cells and the control of T-cell stimulation (Figure 1b). In addition, there are tumor-infiltrating lymphocytes (TILs), in which T cells are removed from inside tumors and “in vitro” expanded to return activated to patients in very high numbers (Figure 1c). Both treatments are promising modalities for advanced cutaneous melanoma, either alone or in combination with other therapies [9,23,24,25,26]. 

Regarding mRNA vaccines, a Phase 3 trial was announced using a personalized vaccine against melanoma, mRNA-4157 (also known as V940), plus the anti-PD-1 drug pembrolizumab (Keytruda) as a combination therapy for high-risk patients who have undergone surgery. Patients who received mRNA-4157 and the anti-PD-1 drug pembrolizumab in the open-label Phase 2 trial showed a 44% decrease in the risk of post-surgical recurrence or death compared with those who received pembrolizumab alone [27].

Other forms of treatment, such as interleukins and oncolytic viruses, also work by stimulating the immune system. Interleukins are proteins that can boost the immune system by inducing T-cell activation. Interleukin-2 (IL-2, aldesleukin, and proleukin) is sometimes used in patients with melanoma and targets T-cell activation, stimulating CD4 Th1 cells, effector cells, and enhancing activation-induced cell death (AICD) [28,29].

Some viruses can be modified in the laboratory to infect and destroy cancer cells. Talimogene laherparepvec, also known as T-VEC, is an oncolytic virus that can be used to treat melanomas and is injected directly into tumors [30]. 

All references used in this review of immunotherapy are listed in Table 1.

### 2.2. Target Therapy 

Targeted therapy is another important approach for the treatment of advanced cutaneous melanoma. This therapy inhibits specific molecules or signaling pathways. The mitogen-activated protein kinase (MAPK) cascade is an intracellular signaling pathway implicated in cell proliferation and survival regulation (Figure 2a) [31,32,33]. Several different mutations, involving BRAF or N-RAS, display an oncogenic effect by activating the MAPK pathway (Figure 2b), resulting in increased tumor cell proliferation [34]. The BRAF mutations are mainly in BRAF-V600E (~80%) and BRAF-V600K (~5 to 30%), which cause alteration of the protein that participates in the MAPK/ERK signaling pathway (Figure 2b,c) [34].

RAS genes (H-RAS, N-RAS, and K-RAS) are mutated and hyperactivated in various cancers. The most common N-RAS mutation is represented by the glutamine to arginine substitution at position 61 (NRASQ61R). Conversely, mutant RAS continues to be a challenging therapeutic target [35]. RAS activation is controlled by GDP and the GTP exchange factor (GEF), and mutant RAS proteins accumulate in the GTP-bound or activated state and are unresponsible to GEF. Several different strategies for directly targeting N-RAS have not resulted in effective therapeutics. However, there have been considerable efforts to develop pharmaceutical strategies aiming to target N-RAS mutations [36]. Selective MEK inhibitors denote a new therapeutic alternative, having been shown to inhibit growth and induce cell death in both BRAF and NRAS mutant melanoma cell lines (Figure 2c,d) [37].

Targeted therapy for cutaneous melanoma is most frequently used in combination with BRAF inhibitors (vemurafenib, dabrafenib, and encorafenib) and MEK inhibitors (trametinib, cobimetinib, and binimetinib). Unfortunately, most patients respond well with an initial response rate (RR) of approximately 53%, but later the response is abrogated due to the development of resistance mechanisms by the tumor. The association of immunotherapy with targeted therapy can be used to overcome resistance but with no expressive results at a longer follow-up [38].

A Phase I trial demonstrated the side effects of anti-MEK. The most common all-grade drug-related adverse events across all dose levels were rash (61.9%), increased creatine phosphokinase (CK, 59.5%), face edema (50.0%), increased aspartate aminotransferase (47.6%), peripheral edema (40.5%), diarrhea (33.3%), alanine aminotransferase (33.3%), and paronychia (19.0%), most of which were grade 1 and 2. The most frequent grade ≥ 3 adverse events were CK (14.2%), asthenia (7.1%), peripheral edema (4.8%), and acneiform dermatitis (4.8%) [39].

In addition, studies have revealed a correlation between melanoma and several somatic driver mutations, including alterations in KIT, a proto-oncogene encoding for a receptor tyrosine kinase (RTK). Although they account for only 3% of all melanomas, mutations in c-KIT are mostly derived from acral, mucosal, and chronically sun-damaged melanomas [40]. Target therapy drugs such as imatinib and nilotinib can block the proliferation of these tumor cells.

Neurotrophic tyrosine receptor kinase (NTRK) fusions are relatively common in spitzoid melanoma, with a prevalence of 21–29% compared to <1% in cutaneous or mucosal melanoma and 2.5% in acral melanoma. The approval of larotrectinib was followed a year later with the approval of another receptor tyrosine kinase (RTK) inhibitor, entrectinib (Rozlytrek), which is indicated for the treatment of adult and adolescent patients with solid tumors harboring NTRK gene fusion and who have no alternative effective therapies available [41]. 

All references used in this review for target therapy are listed in Table 2.

### 2.3. Lysine Histone Methyl Transferase Inhibitors

#### 2.3.1. Methyltransferases

Methyltransferases are essential enzymes for several biological processes, including the regulation of gene expression, modulation of chromatin structure, and cell signaling. These enzymes catalyze the transfer of methyl groups (-CH3) from methyl donors to specific targets such as proteins, nucleic acids, and other molecules. The two most important classes are DNA and protein methyltransferases. 

#### 2.3.2. DNA Methyltransferases

DNA methylation is catalyzed by a group of proteins known as DNA methyltransferases (DNMTs). DMNTs are enzymes that add methyl groups to nitrogenous bases (Figure 3a), which causes a significant impact on the regulation of gene expression since DNA methylation may silence gene activity. The hypermethylation of gene promoter regions can lead to the inhibition of gene transcription and is associated with pathological processes, including cancer [42].

#### 2.3.3. Protein Methyltransferases

Protein methyltransferases (PMT) are enzymes that add methyl groups to amino acid residues and are mainly observed in lysine and arginine residues (Figure 3b). Methylation can affect the structure and function of proteins by regulating enzymatic activity and protein–protein interactions. A well-known example is the methylation of arginine in histones, which are the proteins that help organize DNA into chromatin [43].

#### 2.3.4. Histones and Their Methylation 

Histones are proteins that provide structural support to a chromosome and can undergo a series of post-translational modifications, including acetylation, methylation, phosphorylation, ubiquitination, and sumoylation. 

Histone methylation, analogous to DNA methylation, has been associated with gene repression [44]. However, several lysine methylation patterns can indicate genic expression, such as trimethylated H3K4, or H3K9 monomethylation [45]. 

Histones are methylated by histone methyltransferases (HMT), and methylation is removed by histone demethylases (HDM). 

Methylation of histone occurs either at lysine or arginine residues, on histones H3 and H4 [46]. Methylation of histones H3K4, H3K36, and H3K79 is related to gene activation [46], whereas methylation of H3K9 or H3K27 is correlated with transcriptional repression. On histone H4, K20 methylation is a well-known marker of gene silencing [46]. Similar, to lysine methylation, arginine methylation has been linked to both gene activation (H3R17) and repression (H3R2 and H4R3) [47]. Lysine can be mono-, di-, and trimethylated, whereas arginine can only be mono- or di-methylated [48]. These modifications regulate the ability of transcription factors to access underlying DNA and affect transcription, replication, and chromatin stability [49,50,51]. Histone modifications are critical epigenetic drivers that can alter the chromatin state and are implicated in cancer progression [52].

#### 2.3.5. Histone Methyltransferases (HMTs)

Histone methyltransferases (HMTs) are a group of enzymes that catalyze the addition of methyl groups (-CH3) to specific amino acid residues on histones (Figure 3c). 

HMT is essential for the maintenance of chromatin structure and regulation of gene expression. Dysfunctions in these enzymes can lead to epigenetic changes that are associated with several diseases, including cancer. For example, aberrant histone methylation can result in the inadequate activation or silencing of genes, contributing to the development of tumors [53]. 

#### 2.3.6. Lysine Histone Methyltransferases (KMTs)

Lysine histone methyltransferases (KMTs), also known as histone lysine methyltransferases (HKMTs), are a group of enzymes that catalyze the addition of methyl groups (-CH3) to lysine residues on histones. KMT transfers a methyl group from a methyl donor (usually a coenzyme S-adenosylmethionine, SAM) to a lysine residue present on histones. SAM is assembled with methionine and ATP, which produces phosphate and pyrophosphate. After the transfer of the methyl group, SAM is converted into S-adenonsyl-homocysteine (SHA) (Figure 4a) [54].

The canonical histone lysine methylation is found in humans methylates histones 3 and 4, and specific lysine sites such as H3K4, H3K9, H3K27, H3K36, H3K79, and H4K20. These modifications are generated by 24 different enzymes: 23 SET proteins and one 7βS protein. In general, histone KMTs are highly selective, because the enzymes that methylate H3K36 do not methylate a different lysine if K36 is mutated. Beyond the canonical sites, many other methylation events have been identified by different methods, including mass spectrometry. However, some methylation events are found only during diseases [54].

Canonical histone lysine methylation is associated with different chromatin states and has specific implications for gene regulation. H3K4 is often associated with more accessible and active chromatin regions where gene transcription is more likely to occur. This methylation is generally seen as a marker of active gene enhancers and promoters. H3K9 is typically associated with regions of compact chromatin and gene silencing. This occurs because H3K9 methylation recruits proteins that promote the formation of heterochromatin, a densely packed form of chromatin that tends to silence transcription. H3K27 is also associated with gene silencing. It is often found in the regions of chromatin that regulate cell development and differentiation [55]. 

G9a is a nuclear histone KMT belonging to the Su(var)3-9 family, which mostly catalyzes H3K9 mono- and di-methylation. The G9a SET domain is responsible for the addition of methyl groups on H3, whereas the ankyrin repeats have been described to denote mono- and di-methyl lysine binding regions [56]. Thus, G9a is not only able to methylate histone tails but also identify this modification, operating as a platform for the recruitment of other target molecules on the chromatin [57].

A G9a-like protein (GLP) interacts with G9a, forming a heterodimeric complex. This heteromeric structure (G9a–GLP) is the main form and represents the methyltransferase in vivo active status [58]. Although the heterodimer appears to be essential for G9a–GLP methyltransferase activity, the enzymatic activity of G9a is more important for the in vivo function of the complex [44].

H3K9 is initially methylated by G9a–GLP to form active H3K9 mono-methylation, which is subsequently methylated by G9a to form repressive H3K9 di-methylation (H3K9me2). G9a directly contributes to the di-methylation and subsequent tri-methylation of H3K9 across the genome (Figure 4b). Depending on the modified residue position, histone methylation can either suppress (H3K9 and H3K27) or enhance (H3K4) gene expression [55].

During hypoxia, histone methyltransferase G9a activity increases, leading to a global increase in histone H3K9 methylation (Figure 4c). This higher methylation inhibits the expression of cell adhesion molecules such as E-cadherin. The correlation between the G9a-mediated repression of cell adhesion molecules and their increased activity during hypoxia strongly supports the direct involvement of G9a in the metastatic pathway [55].

In addition, G9-enhanced expression has been linked to the malignant behaviors of cancer cells such as aberrant proliferation, metastasis, and drug resistance by silencing tumor suppressors or by activating epithelia–mesenchymal transition programs [59,60]. Its overexpression is associated with different types of cancer and poor prognosis, including melanoma, due to the positive regulation of the Nocth1 signaling pathway [56,61,62]. Enhanced signaling from the Notch1 pathway specifically contributes to the development of melanoma, allowing the survival and proliferation of these cells in stressful and hypoxic environments [63,64,65,66].

Consequently, lysine methyltransferase (KMT) can be a target molecule for cancer and metastasis treatment; however, different from the therapies discussed before, these therapies are under development. The first FDA approval for a KMT inhibitor (tazemetostat for epithelioid sarcoma 4 and subsequently follicular lymphoma) occurred in 2020, and, up to now, no other current approvals are known [67].

#### 2.3.7. KMT Inhibitors

KMT inhibitors are compounds that target KMT enzymes. These inhibitors were developed to modulate the activity of KMT and, consequently, the epigenetic modifications of histones. This may have implications for the regulation of gene expression and associated cellular processes, including development, cell differentiation, and disease [68].

KMT inhibitors can be classified into two main types:-Selective inhibitors: These inhibitors target a specific KMT enzyme. This allows a precise approach by targeting specific epigenetic modifications at specific histone sites. Examples of selective inhibitors include those that target KMT for H3K9 or H3K27 methylation [67].-Broad-spectrum inhibitors: These inhibitors target multiple KMTs in a less specific manner. They affect a broader spectrum of epigenetic modifications and may have widespread effects on cells. This can be beneficial for manipulating gene expression, but may also result in off-targeting side effects [67]. Research into KMT inhibitors is an expanding area with significant implications in basic research and the development of potential new drugs. However, the complexity of epigenetic regulation, and the interconnection of cell signaling pathways, is a challenge. Furthermore, specificity is a critical aspect because the inappropriate inhibition of a KMT can lead to off-target side effects.

#### 2.3.8. G9a–GLP Inhibitors

G9a and GLP have approximately 80% sequence identity in their conserved catalytic SET domains, which establishes a challenge for the development of dual selective inhibitors [69]. Several G9a–GLP dual inhibitors target either the SAM binding site (methyl donor) or the substrate binding pocket [70].

Currently, three compounds are identified as G9a–GLP inhibitors. The compounds BIX01294 and UNC0638 are being tested in vitro, and UNC0642 is being tested in vivo in preclinical studies with animal models.

In this sense, the compound UNC0642 acts as a competitive inhibitor of G9a activity, connecting to its binding pocket and making it inaccessible (Figure 5a). This compound is described as a potent inhibitor of G9a–GLP with low toxicity and high selectivity [71,72].

UNC0642 studies are more advanced because of their favorable pharmacokinetics, better half-life, high selectivity, and low cellular toxicity, making them more suitable for preclinical studies [72,73].

G9a–GLP is implicated in several mechanisms and pathways, including inhibition of the expression of the caspase 1 gene (CASP1) by enhancing the H3K9me2 of the promoter region. In addition, G9a can di-methylate p53, inhibiting its activity and enhancing the expression of polo-like kinase 1 (Plk1), which has been implicated in cell proliferation and epithelial–mesenchymal transition (Figure 5b) [74].

Consequently, G9a–GLP inhibitors can impede tumor growth by blocking p53 di-methylation. Active p53 will block the cell cycle progression of cells with damage in their DNA and reduce Plk1 expression. In addition, the unblocking of caspase 1 expression will support the cells’ inflammasome formation, allowing tumor cell death [60,74].

With regard to melanoma, as discussed before, Noch-1 signaling is enhanced in the disease, and also by H3K9 di-trimethylation, which leads to MEK inhibitor resistance [75] and microphthalmia-associated transcription factor (MITF) repression via a competition-based mechanism, thereby triggering critical transit into the invasive melanoma stage [76]. Thus, G9a inhibitors, in addition to impairing common cancer mechanisms, also impair specific mechanisms related to melanoma [61].

G9a plays a significant role in cancer processes such as heterochromatin formation, DNA methylation, transcriptional silencing, proliferation, cell death, differentiation, and the mobility of tumor cells [77]. Thus, G9a is a novel therapeutic target for anticancer agents and the development of novel G9a inhibitors may provide a new option for invasive melanoma treatment. Although the knowledge of the mechanisms involved in KMT inhibitors is being improved, much remains to be answered, and, in this sense, experimental research and preclinical studies still have plenty to add to the current knowledge.

A recent preclinical study combining a G9a inhibitor (UNC0642) with an anti-PD1 antibody showed increased immunotherapy efficacy, with increased survival and a lower incidence of acquired resistance to checkpoint inhibitors in murine melanoma [78]. Clinical studies using KMT inhibitors for advanced cutaneous melanoma are still in the early stages. They are being tested alone or in combination with immune checkpoint blockade (anti-CTLA4 and anti-PD1) or targeted therapy agents (BRAF/MEK inhibitors).

It is important to note that many studies are exploring the therapeutic implications of G9a inhibitors and their possible side effects. Importantly, as with any pharmacological drug development, safety, efficacy, and specificity are crucial considerations when developing new drugs for clinical applications.

All references used in this review for histone methyltransferases are listed in Table 3.

## 3. Nicotinamide N-Methyltransferase

Nicotinamide N-methyltransferase (NNMT) is a cytosolic enzyme that catalyzes the conversion of nicotinamide (NA, vitamin b3) to 1-methylnicotinamide (MNA), using S-adenosyl-L-methionine (SAM) as a methyl donor, and plays an important role in drug metabolism. NNMT is mainly expressed in the liver and adipose tissue. One of the primary roles of NNMT is the detoxification of xenobiotics. The function of NNMT is the recognition of a substrate that allows for the methylation of different metabolites, including pyridines, quinolines, and other related heterocyclic aromatics. Its upregulation has been reported in several neoplasms contributing to chemotherapy drug resistance [79].

Ganzetti’s study performed an immunohistochemical evaluation of NNMT expression in tissue samples obtained from patients with melanoma using nevi specimens as controls. They demonstrated that cutaneous melanoma showed a significantly higher NNMT expression than benign nevi [80]. A subsequent study demonstrated the presence of NNMT in metastatic lymph nodes [81].

Van Haren et al. established efforts focusing on the potential of using prodrug strategies for the delivery of polar NNMT inhibitors into cells [82]. Following this study, another study from the same author isolated and tested five macrocyclic peptides, which showed the potent inhibition of NNMT.

In addition, Gao et al. reported strategies to generate novel and potent NNMT inhibitors, including a nonbenzamide aromatic-mimicking nicotinamide group and employing a three-carbon trans-alkene linker to connect these aromatic groups to the SAM unit [83].

Even in very preliminary studies, these compounds appear to impair NNMT activity and reduce drug resistance, which should be further investigated in pre-clinical studies.

All references used in this review for nicotinamide N-methyltransferase are listed in Table 4.

## 4. Conclusions

Most patients with advanced cutaneous melanoma usually die of the disease. Thus, new treatments for these patients are necessary. A line in the search for new treatments is epigenetic medicines. Lysine methyl transferase inhibitors are being tested alone or in combination with immune checkpoint inhibitors or target therapy in pre-clinical studies and have demonstrated the potential to reduce the acquired resistance to these treatments. However, even though there are promising new therapies, their clinical effectiveness and performance will only be demonstrated after clinical trials are completed.

## Figures and Tables

**Figure 1 cancers-15-05751-f001:**
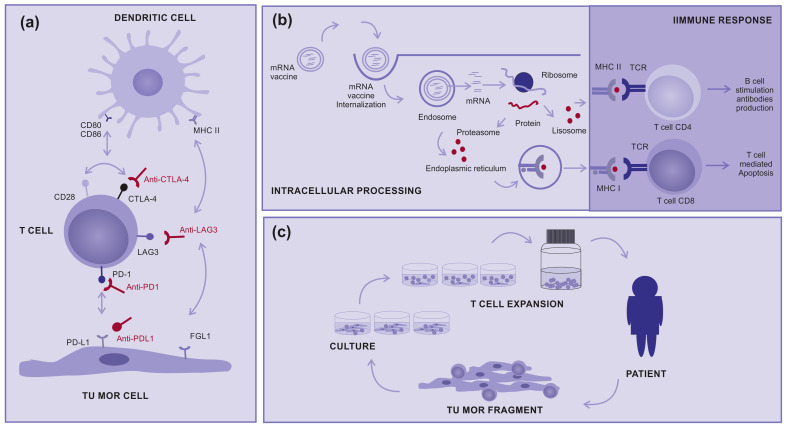
**How immunotherapy works:** (**a**) Immune checkpoint inhibitors: Molecules designed to block immune checkpoints (ex: CTLA4, PD-1, PD-L1, or LAG3) interact with specific ligands, (**b**) mRNA vaccine: Encapsulated mRNA targets the immune cells, leading to protein translation intended for their processing and presentation to T cells via MHC class I and II, aiming for a subsequent immune response, and (**c**) Tumor-infiltrating Lymphocytes (TILs): A tumor fragment is removed from the patient during surgery and cultured in the laboratory to a logarithmic expansion of TILs, with the intention to return them to the patient.

**Figure 2 cancers-15-05751-f002:**
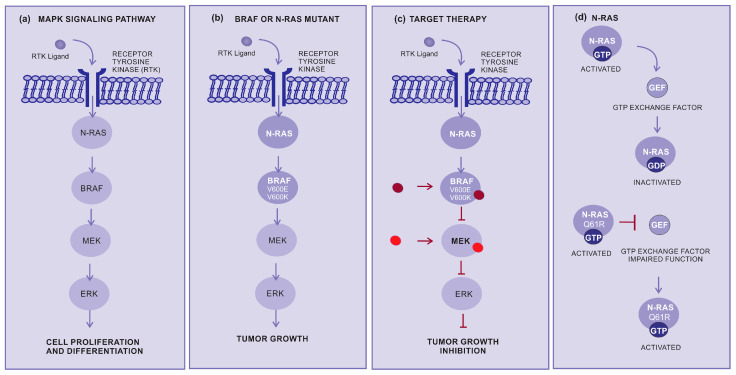
How target therapy works: (**a**) MAPK pathway normal signaling: The BRAF and MEK pathways initiate signaling through the extracellular RTK domain. RAS family members are activated through an RTK ligand domain, and the activation of RAS proteins binds BRAF isoforms, leading to the activation of BRAF, followed by MEK, and the final step of phosphorylation of ERK. The net effect leads to increased cell survival and decreased apoptosis. (**b**) BRAF or N-RAS mutant pathway: In the presence of an activating BRAFV600E mutation, BRAF no longer requires dimerization with RAS, and therefore remains constitutively active, similar to NRAS mutations, keeping this pathway constitutively activated and leading to enhanced tumor cell proliferation with apoptosis abrogation. (**c**) Inhibition of the BRAF or N-RAS mutant pathway: The targeted therapy inhibits mutant BRAF or MEK, thereby stopping the downstream activation of the MAPK pathway, decreasing cellular proliferation, and inducing apoptosis. (**d**) N-RAS: The N-RAS mutation turns the protein that is constantly activated (GTP bound) and irresponsive to the GTP exchange factor. Abbreviations: ERK, extracellular signal-regulated kinase; MAPK, mitogen-activated protein kinase; BRAF, rapidly accelerated fibrosarcoma B type; RAS, rat sarcoma virus; and RTK, receptor tyrosine kinases.

**Figure 3 cancers-15-05751-f003:**
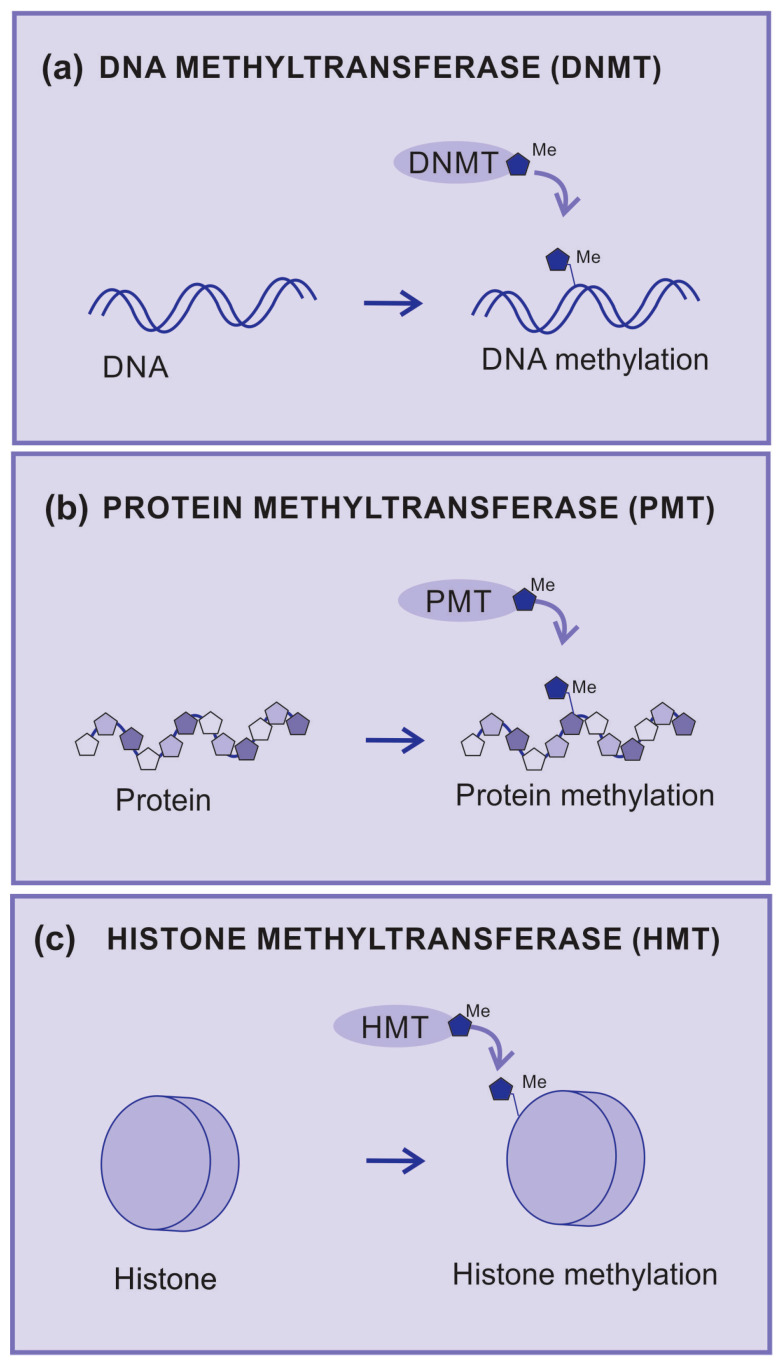
Methyltransferases. (**a**) DNA methyltransferase (DNMT) is an enzyme that transfers radical methyl to DNA nucleotides, (**b**) protein methyltransferase (PMT) is an enzyme that transfers radical methyl to protein amino acids, (**c**) histone methyltransferase (HMT) is an enzyme that transfers radical methyl to histones.

**Figure 4 cancers-15-05751-f004:**
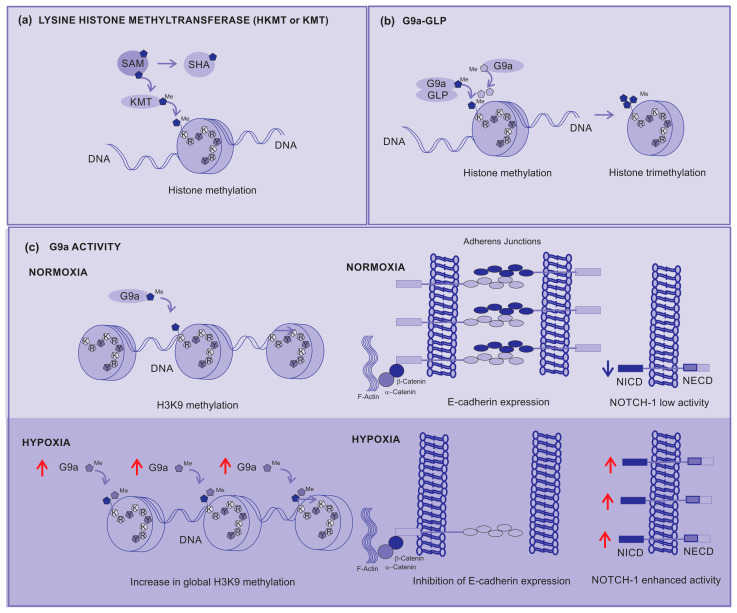
Lysine Histone Methyltransferases (KMTs) (**a**) KMT transfers a methyl from a donor (SAM) to the histone, (**b**) the G9a-GLP complex is responsible for the mono-methylation of H3K9 histone, but G9a is responsible for the di- and tri-methylation of H3K9. (**c**) G9a activity under normoxia contributes to the prompt di-methylation of H3K9, but under hypoxia it increases the global di-methylation of H3K9, silencing the gene responsible for E-cadherin expression, affecting the adhesion between cells, contributing to metastasis, and activating the Notch-1 signaling pathway which contributes to cancer cell survival and proliferation. Red arrows—up arrow is an indicative of upregulation.

**Figure 5 cancers-15-05751-f005:**
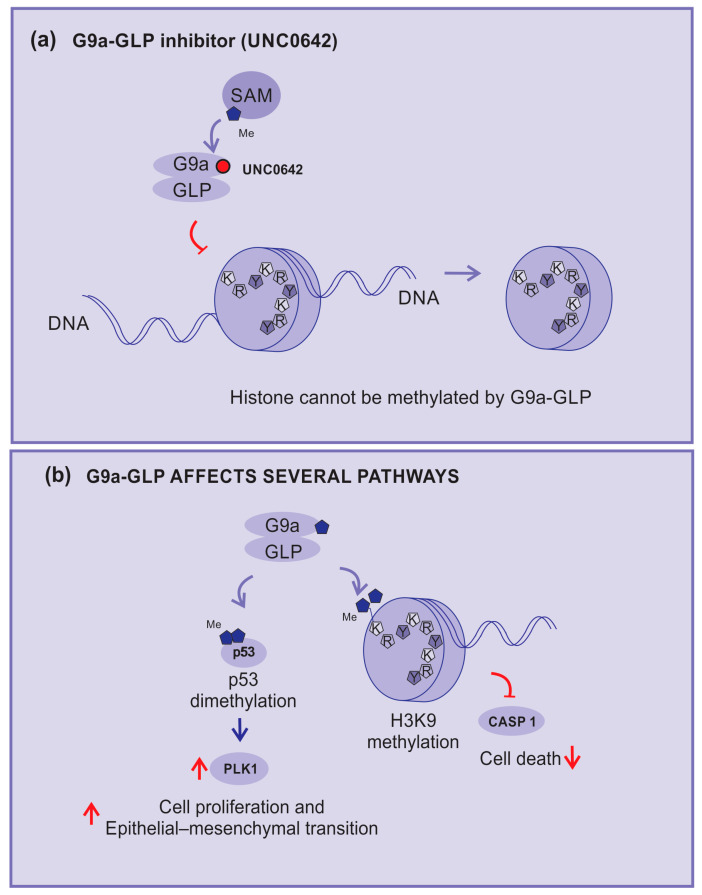
KMT inhibitors. (**a**) UNC0642: an example of methylation inhibition by competitive blocking of the G9a binding pocket, (**b**) G9a and GLP-affected pathways: H3K9 di-methylation inhibits caspase 1 expression; G9a demethylates p53, inhibits its activity, and enhances the expression of polo-like kinase 1 (PLK1), which is implicated in cell proliferation and epithelial–mesenchymal transition (EMT). Red arrows—up arrow is an indicative of upregulation, down arrow is an indicative of downregulation, and blunt arrow is an indicative of impairment.

**Table 1 cancers-15-05751-t001:** Main characteristics of immunotherapy studies.

Reference	Journal	Study Subject	Study Type
Qin and Zeng, 2023 [9]	*Exp. Ther. Med.*	Targeted therapy and immunotherapy for melanoma	Review Article
Hold et al., 2010 [10]	*N. Engl. J. Med.*	Ipilimumab and patient suvival in melanoma	Original Article (Clinical trial)
Eggermont et al., 2016 [11]	*N. Engl. J. Med.*	Ipilimumab and patient suvival in melanoma	Original Article (Clinical trial)
Weber et al., 2017 [12]	*N. Engl. J. Med*.	Nivolumab versus ipilimumab in resected Stage III or IV melanoma	Original Article (Clinical trial)
Tawbi et al., 2022 [13]	*N. Engl. J. Med.*	Relatlimab and nivolumab versus nivolumab in untreated advanced melanoma	Original Article (Clinical trial)
Zhang et al., 2021 [14]	*J. Exp. Clin. Cancer*	Regulatory mechanisms of immune checkpoints PD-L1 and CTLA-4 in cancer	Review Article
Maruhashi et al., 2020 [15]	*J. Immunother. Cancer*	LAG-3 molecular functions and clinical applications	Review Article
Bai et al., 2020 [16]	*Front. Oncol.*	Mechanisms of cancer resistance to immunotherapy	Review Article
Ribas et al., 2005 [17]	*J. Clin. Oncol.*	Phase I trial with CTLA4 in melanoma	Original Article
Liao et al., 2014 [18]	*Neuro Oncol.*	Neurological complications related to ipimumab	Case Report
Bertrand et al., 2015 [19]	*BMC Med.*	Meta-analysis on related adverse events associated with anti-CTLA-4 antibodies	Review Article
Anderson et al., 2016 [20]	*Immunity*	Immune function of Lag-3, Tim-3, and TIGIT	Review Article
Chibara et al., 2018 [21]	*Nature*	Transcriptional regulation of co-receptors PD-1, Tim-3, Lag-3, and TIGIT and description of new ones	Original Article
Wei and Li, 2022 [22]	*Front. Oncol.*	Drug synergy of PD-1 and LAG-3 and drug resistance to overcome	Original Article
Tawbi et al., 2022 [13]	*N. Engl. J. Med.*	Relatlimab and nivolumab versus nivolumab in untreated advanced melanoma	Original Article
Rosenberg et al., 2011 [23]	*Clin. Cancer Res.*	T-cell transfer immunotherapy and patient response	Original Article
Sarnaik et al., 2021 [24]	*J. Clin. Oncol.*	Lifileucel, a tumor-infiltrating lymphocyte therapy, in metastatic melanoma	Original Article
Seitter et al., 2021 [25]	*Clin. Cancer Res.*	Adoptive transfer of tumor-infiltrating lymphocytes in patients with metastatic melanoma	Original Article
Qin et al., 2022 [26]	*Signal Transduct Target Ther*.	mRNA-based therapeutics	Review Article
Carvalho T, 2023 [27]	*Nat. Med.*	mRNA and immunotherapy tested in melanoma trial	Original Article
Rosenberg et al., 1985 [28]	*N. Engl. J. Med.*	T-cell transfer immunotherapy and IL-2 in melanoma	Original Article
Muhammad et al., 2023 [29]	*Mol. Cancer*	IL-2 and IL-2R targeting strategies	Review Article
Andtbacka et al., 2019 [30]	*J. Immunother. Cancer*	Talimogene laherparepvec litic virus in melanoma	Original Article

**Table 2 cancers-15-05751-t002:** Main characteristics of target therapy studies.

Reference	Journal	Study Subject	Study Type
Chang et al., 2001 [31]	*Nature*	MAPK signaling	Review Article
Liebmann et al., 2001 [32]	*Cell Signal*	MAPK signaling	Review Article
Grimaldi et al., 2014 [33]	*Curr. Opin. Oncol.*	MEK inhibitors in metastatic melanoma	Review Article
Thompson and Lyons, 2005 [34]	*Curr. Opin. Pharmacol.*	Advances in MEK pathway inhibitors in cancer	Review Article
Sexton et al., 2019 [35]	*Semin. Cancer Biol.*	RAS and exosome signaling	Review Article
Papke and Der, 2017 [36]	*Science*	Drugs targting RAS	Review Article
Grimaldi et al., 2017 [37]	*Am. J. Clin. Dermatol.*	MEK inhibitors in the treatment of metastatic melanoma	Review Article
Lopes et al., 2022 [38]	*Cancers*	Melanoma epidemiology, treatment, and latest advances	Review Article
Wang et al., 2023 [39]	*BMC Med.*	Phase I dose-escalation and dose-expansion trial of a selective MEK inhibitor	Original Article (Clinical trial)
Pham and Tsao, 2023 [40]	*Yonsei Med. J.*	KIT and melanoma: biological insights and clinical implications	Review Article
Forschner et al., 2020 [41]	*J. Dtsch. Dermatol. Ges.*	NTRK gene fusions in melanoma and potential therapeutic implications	Review Article

**Table 3 cancers-15-05751-t003:** Main characteristics of histone methyltransferase studies.

Reference	Journal	Study Subject	Study Type
Del Castilho et al., 2022 [42]	*Int. J. Mol. Sci.*	DNA methyltransferases evolution to clinical applications	Review Article
Kaniskan et al., 2018 [43]	*Chem. Rev.*	Inhibitors of protein methyltransferases and demethylases	Review Article
Tachibana et al., 2008 [44]	*EMBO*	G9a/GLP complexes mediate H3K9 and DNA methylation	Original Article
Barski et al., 2007 [45]	*Cell*	High-resolution profiling of histone methylations in the human genome	Original Article
Black et al., 2012 [46]	*Mol. Cell.*	Histone lysine methylation dynamics	Review Article
Zhao et al., 2009 [47]	*Nat. Struct. Mol. Biol.*	Role of PRMT5 in DNA methylation and gene silencing	Original Article
Rea et al., 2000 [48]	*Nature*	Regulation of chromatin structure by histone H3 methyltransferases	Original Article
Bates, 2021 [49]	*N. Engl. J. Med.*	Epigenetic therapies for cancer	Review Article
Moran et al., 2021 [50]	*Semin. Cancer Biol.*	Epigenetics of malignant melanoma	Review Article
Harel and lupski, 2021 [51]	*Clin. Genet.*	Mechanisms on genomic disorders and clinical manifestations	Review Article
Sang and deng, 2019 [52]	*Dermatol. Ther.*	Epigenetic mechanisms of skin cancer	Review Article
Nacev et al., 2019 [53]	*Nature*	Oncohistone mutations in human cancers	Original Article
Husmann and Gozani, 2019 [54]	*Nat. Struct. Mol. Biol.*	Histone lysine methyltransferase biology	Review Article
Karami et al., 2022 [55]	*Cancer Cell Int.*	Epigenetics role in melanoma treatment and resistance	Review Article
Casciello et al., 2015 [56]	*Front. Immunol.*	G9a histone methyltransferase in cancer	Review Article
Shahbazian et al., 2005 [57]	*Mol. Cell.*	Histone ubiquitylation controls processive methylation	Original Article
Tachibana et al., 2005 [58]	*Genes Dev.*	Histone methyltransferases G9a and GLP	Original Article
Fan et al., 2015 [59]	*Genome Biol.*	Gene-specific histone methylation can promote tumorigenesis	Original Article
Liao et al., 2023 [60]	*J. Pharm. Anal.*	Histone lysine methyltransferases in cancer therapy	Review Article
Dang et al., 2020 [61]	*Aging*	G9a in melanoma cells promotes upregulation of the Notch1 signaling pathway	Original Article
Filho et al., 2021 [62]	*Braz. J. Nat. Sci.*	Notch receptors as a therapeutic target in melanoma	Review Article
Ayaz and Osborne, 2014 [63]	*Front. Oncol.*	Non-canonical Notch signaling in cancer and immunity	Review Article
Bedogni, 2014 [64]	*Pigment Cell Mel. Res.*	Notch signaling in melanoma	Review Article
Zhang et al., 2016 [65]	*J. Investig. Dermatol.*	Melanoma through inhibition of Notch1 and ERBB3	Original Article
Tang et al., 2019 [66]	*Neoplasma*	EGFL7 silencing inactivates the Notch signaling pathway in human cutaneous melanoma	Original Article
Bhat et al., 2021 [67]	*Nat. Rev. Drug Discov.*	Epigenetics and protein lysine methylation to treat disease	Review Article
Rugo et al., 2020 [68]	*Adv. Ther.*	Histone methyltransferase inhibitors in clinical oncology	Review Article
Link et al., 2009 [69]	*Mol. Cancer Res.*	Histone methyltransferases G9a and GLP in cancer germ-line antigen gene regulation	Original Article
Sweis et al., 2014 [70]	*ACS Med. Chem. Lett.*	Discovery of potent and selective inhibitors of histone methyltransferase G9a	Original Article
Park et al., 2022 [71]	*J. Med. Chem.*	Discovery of G9a/GLP covalent inhibitors	Original Article
Liu et al., 2013 [72]	*J. Med. Chem.*	Discovery of an in vivo chemical probe of the lysine methyltransferases G9a and GLP	Original Article
Flesher and Fisher, 2019 [73]	*Eur. J. Med. Chem.*	Progress in histone methyltransferase (G9a) inhibitors	Review Article
Cao et al., 2019 [74]	*Eur. J. Med. Chem.*	Progress in histone methyltransferase (G9a) inhibitors	Review Article
Porcelli et al., 2021 [75]	*Biomed Pharmacother.*	Notch protects MAPK-activated melanoma from MEK inhibitor	Review Article
Golan and Levy, 2019 [76]	*Int. J. Mol. Sci.*	Microphthalmia-Associated Transcription Factor (MITF) and Notch signaling	Original Article
Kato et al., 2020 [77]	*Cancer Discov.*	Gain-of-function genetic alterations of G9a drive oncogenesis	Original Article
Kelly et al., 2021 [78]	*Clin. Cancer Res.*	G9a inhibition enhances checkpoint inhibitor blockade response in melanoma	Original Article

**Table 4 cancers-15-05751-t004:** Main characteristics of nicotinamide N-Methyltransferase studies.

Reference	Journal	Study Subject	Study Type
Pompei et al., 2019 [79]	*Eur. J. Clin. Investig.*	Nicotinamide N-methyltransferase in nonmelanoma skin cancers	Original Article
Ganzetti et al., 2018 [80]	*Melanoma Res.*	Nicotinamide N-methyltransferase presence in cutaneous malignant melanoma.	Original Article
Sartini et al., 2023 [81]	*Hum. Cell.*	Nicotinamide N-methyltransferase presence in melanoma lymph node metastasis	Original Article
van Haren et al., 2021 [82]	*RSC Chem. Biol.*	Macrocyclic peptides as allosteric inhibitors of nicotinamide N-methyltransferase	Original Article
Gao et al., 2012 [83]	*J. Med. Chem.*	Potent inhibition of nicotinamide N-Methyltransferase by alkene-linked bisubstrate	Original Article

## Data Availability

The data can be shared up on request.

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
