# Peer review of "A Review of Advanced Cutaneous Melanoma Therapies and Their Mechanisms, from Immunotherapies to Lysine Histone Methyl Transferase Inhibitors"

_cancers, 2023, doi:10.3390/cancers15245751_

Round 1

Reviewer 1 Report

Comments and Suggestions for Authors

The manuscript “A review for advanced cutaneous melanoma therapies and their mechanisms, from immunotherapies to lysine histone methyl transferase inhibitors” is a review article regarding the current options for the management of advanced cutaneous melanoma.

Although the manuscript might be of interest for the readers, it covers only partially the current available literature on this topic. Therefore, as it is, the manuscript cannot be accepted for publication.

Authors are strongly encouraged to improve the manuscript addressing the following concerns:

1.       An accurate revision of the manuscript regarding the English language is mandatory; there are many mistakes and typos; e.g. “The canonical histone lysine methylation is associated with different chromatin states and have specific implications in gene regulation” should be “and HAS specific implications in gene regulation”

2.       The introduction section should include more information about advanced cutaneous melanoma, underlying also the clinical difficulty in the management of this malignancy. A proper background will help the reader to focus on the clinical problem.

3.       In the introduction section references are not in square brackets.

4.       Figure 1 contains few mistakes: the word Immune is written IIMUNE in the upper right corner. Expantion istead of expansion.

5.       Figure 2 contains mistake: inhibtion instead of inhibition.

6.       The main concern of this review is that this covers only partially the literature available. For instance, the enzyme nicotinamide N-methyltransferase (NNMT) has been reported to be overexpressed in a number of solid malignancies including melanoma, where it contributes to the tumorigenicity and aggressiveness. It was reported that melanoma displays high NNMT expression compared to nevi (PMID: 29420365). Moreover, lymph node metastases from cutaneous melanoma displays higher positivity to NNMT expression compared to primary melanomas and nevi, and NNMT enhances the chemoresistance of melanoma cells. For all these reasons, NNMT is an excellent target in advanced cutaneous melanoma (PMID: 36151433).

A number of NNMT inhibitors are already available, and their use is a promising strategy for targeted therapy in cancer (PMID: 34572571; PMID: 34704059; PMID: 34424711).

7.       Line 214: the sentence “Beyond the canonical sites, 214 many other methylation events have been identified by different methods, including mass 215 spectrometry, some only are found during diseases or )” is cut and has no sense.

8.       Figure 4 contains mistake: inhibtion instead of inhibition.

9.       Figure 5 contains mistake: “affect” must be “affects”; can not should be cannot.

Comments on the Quality of English Language

Extensive editing of English language required

Author Response

Response to reviewers

  1. An accurate revision of the manuscript regarding the English language is mandatory; there are many mistakes and typos; e.g. “The canonical histone lysine methylation is associated with different chromatin states and have specific implications in gene regulation” should be “and HAS specific implications in gene regulation”

Answer: The article was entirely revised

  1. The introduction section should include more information about advanced cutaneous melanoma, underlying also, the clinical difficulty in the management of this malignancy. A proper background will help the reader to focus on the clinical problem.

Answer: The introduction was modified to present more information about the advanced melanoma and the difficult in its treatment.

  1. In the introduction section references are not in square brackets.

Answer: Thank you for referring that, this typo was correct.

  1. Figure 1 contains few mistakes: the word Immune is written IIMUNE in the upper right corner. Expantion istead of expansion.

Answer: These corrections were performed.

  1. Figure 2 contains mistake: inhibtion instead of inhibition.

Answer: This mistake was corrected

  1. The main concern of this review is that this covers only partially the literature available. For instance, the enzyme nicotinamide N-methyltransferase (NNMT) has been reported to be overexpressed in a number of solid malignancies including melanoma, where it contributes to the tumorigenicity and aggressiveness. It was reported that melanoma displays high NNMT expression compared to nevi (PMID: 29420365). Moreover, lymph node metastases from cutaneous melanoma displays higher positivity to NNMT expression compared to primary melanomas and nevi, and NNMT enhances the chemoresistance of melanoma cells. For all these reasons, NNMT is an excellent target in advanced cutaneous melanoma (PMID: 36151433). A number of NNMT inhibitors are already available, and their use is a promising strategy for targeted therapy in cancer (PMID: 34572571; PMID: 34704059; PMID: 34424711).

Answer: A new topic on NNMT was included and the following references were added to the manuscript:

  1. Pompei V, Salvolini E, Rubini C, Lucarini G, Molinelli E, Brisigotti V, Pozzi V, Sartini D, Campanati A, Offidani A, Emanuelli M. Nicotinamide N-methyltransferase in nonmelanoma skin cancers. Eur J Clin Invest. 2019; 49(12):e13175. doi: 10.1111/eci.13175. 
  2. Ganzetti G, Sartini D, Campanati A, Rubini C, Molinelli E, Brisigotti V, Cecati M, Pozzi V, Campagna R, Offidani A, Emanuelli M. Nicotinamide N-methyltransferase: potential involvement in cutaneous malignant melanoma. Melanoma Res. 2018;28(2):82-88. doi: 10.1097/CMR.0000000000000430. 
  3. Sartini D, Molinelli E, Pozzi V, Campagna R, Salvolini E, Rubini C, Goteri G, Simonetti O, Campanati A, Offidani A, Emanuelli M. Immunohistochemical expression of nicotinamide N-methyltransferase in lymph node metastases from cutaneous malignant melanoma. Hum Cell. 2023;36(1):480-482. doi: 10.1007/s13577-022-00793-3. 
  4. van Haren MJ, Zhang Y, Thijssen V, Buijs N, Gao Y, Mateuszuk L, Fedak FA, Kij A, Campagna R, Sartini D, Emanuelli M, Chlopicki S, Jongkees SAK, Martin NI. Macrocyclic peptides as allosteric inhibitors of nicotinamide N-methyltransferase (NNMT). RSC Chem Biol. 2021;2(5):1546-1555. doi: 10.1039/d1cb00134e. 
  5. Gao Y, van Haren MJ, Buijs N, Innocenti P, Zhang Y, Sartini D, Campagna R, Emanuelli M, Parsons RB, Jespers W, Gutiérrez-de-Terán H, van Westen GJP, Martin NI. Potent Inhibition of Nicotinamide N-Methyltransferase by Alkene-Linked Bisubstrate Mimics Bearing Electron Deficient Aromatics. J Med Chem. 2021;64(17):12938-12963. doi: 10.1021/acs.jmedchem.1c01094.
  6. Line 214: the sentence “Beyond the canonical sites, 214 many other methylation events have been identified by different methods, including mass 215 spectrometry, some only are found during diseases or)” is cut and has no sense.

Answer: This phrase was corrected

  1. Figure 4 contains mistake: inhibtion instead of inhibition.

Answer: This figure was corrected

  1. Figure 5 contains mistake: “affect” must be “affects”; can not should be cannot.

Answer: This figure was corrected

Reviewer 2 Report

Comments and Suggestions for Authors

In this review, Oliveira Filho and colleagues reported an overview of the current knowledge about recent available therapies and the ones in development for advanced cutaneous melanoma therapy. They described Advanced Melanoma Treatments such as immunotherapy and Target therapy. Then they focused on Lysine histone methyl transferase inhibitors classes, such as Methyltransferases, DNA methyltransferases, Protein methyltransferases, Histones and Histone methylation, Histone Methyltransferases (HMT) and Lysine Histone Methyltransferases (KMT). After the description of their mechanism, authors listed the classes of main KMT inhibitors, paying attention to their potential applications in the clinic. The topic covered is very common so this review is not very innovative compared to the numerous reviews in the field.

The review is clear and concise, and the references are appropriate. The figures are explanatory of the content. However, the authors are suggested to introduce tables that summarize the main publications and clinical studies associated with the various paragraphs.

Author Response

The review is clear and concise. The figures are explanatory of the content. However, the authors are suggested to introduce tables that summarize the main publications and clinical studies associated with the various paragraphs

Answer: We have included Tables at the end of each topic, summarizing the publications.

Reviewer 3 Report

Comments and Suggestions for Authors

The paper should be according to the title: "A review for advanced cutaneous melanoma therapies and their mechanisms, from immunotherapies to lysine histone methyl transferase inhibitors".

I found a great imbalance considering evidence derived from  RCTs and RWTs on the limited discussion on immunotherapies and targeted therapies versus Lysine Histone Methyltransferases inhibitors, considering that the KMTis, despite their proven efficacy in hematologic neoplasms, are not yet proven effective in metastatic melanoma and the majority of  RCTs have been discontinued due to futility. I would dissect more extensively the new combinations of  PD1 antibodies with anti-LAG-3,  with vaccines, or with the new bifunctional antibodies.

Regarding targeted therapies, I would discuss the lack of effective inhibitors for NRAS mutations, and the limited effect and not neglectable toxicity of anti MEKi in NRAS mut melanoma.

Considering c-KITis, they showed efficacy only in the presence of exon 11 and 13 mutations and usually with short clinical effects. I would not forget the occurrence of  NTRK fusion mutations in some types of melanoma that could be effectively treated with entrectinib or larotrectinib.

Finally, I do not agree with the statement on line 133 "The association of immunotherapy with targeted therapy is widely used to overcome resistance", since the only combination approved  is  atezolizumab + cobimetinib and vemurafenib which  resulted more effective than vem-cobi alone in the first analysis, but recently, at a longer follow-up was reported as not more effective on survival than combi + vemu alone.  Furthermore, the associations of anti PD1  and targeted therapies are less effective than ipi-nivo combinations and more toxic.

Comments on the Quality of English Language

In the text there are typing errors  that should be checked 

Author Response

The paper should be according to the title: "A review for advanced cutaneous melanoma therapies and their mechanisms, from immunotherapies to lysine histone methyl transferase inhibitors".

I found a great imbalance considering evidence derived from  RCTs and RWTs on the limited discussion on immunotherapies and targeted therapies versus Lysine Histone Methyltransferases inhibitors, considering that the KMTis, despite their proven efficacy in hematologic neoplasms, are not yet proven effective in metastatic melanoma and the majority of  RCTs have been discontinued due to futility. I would dissect more extensively the new combinations of PD1 antibodies with anti-LAG-3, with vaccines, or with the new bifunctional antibodies.

Answer:

1- Regarding PD1 and LAG3, we have included a trial result about the combination of anti-PD-1 and LAG3 and the 2 references listed below:

- Wei Y, Li Z. LAG3-PD-1 Combo Overcome the Disadvantage of Drug Resistance. Front Oncol. 2022;12:831407. doi: 10.3389/fonc.2022.831407. 

- Tawbi HA, Schadendorf D, Lipson EJ, Ascierto PA, Matamala L, Castillo Gutiérrez E, Rutkowski P, Gogas HJ, Lao CD, De Menezes JJ, Dalle S, Arance A, Grob JJ, Srivastava S, Abaskharoun M, Hamilton M, Keidel S, Simonsen KL, Sobiesk AM, Li B, Hodi FS, Long GV; RELATIVITY-047 Investigators. Relatlimab and Nivolumab versus Nivolumab in Untreated Advanced Melanoma. N Engl J Med. 2022 Jan 6;386(1):24-34. doi: 10.1056/NEJMoa2109970. 

2- Regarding PD1 and RNA vaccines, we have included a melanoma trial that showed a decrease of 44% in the risk of post-surgical recurrence or death compared to pembrolizumab alone, and the reference listed below:

- Carvalho T. Personalized anti-cancer vaccine combining mRNA and immunotherapy tested in melanoma trial. Nat Med. 2023 Oct;29(10):2379-2380. doi: 10.1038/d41591-023-00072-0. 

Regarding targeted therapies, I would discuss the lack of effective inhibitors for NRAS mutations, and the limited effect and not neglectable toxicity of anti MEK and NRAS mut melanoma.

Answer: A discussion of the lack of NRAS mutations inhibitors was included. The target therapy limited, and side effect was also discussed.

Considering c-KITis, they showed efficacy only in the presence of exon 11 and 13 mutations and usually with short clinical effects. I would not forget the occurrence of  NTRK fusion mutations in some types of melanoma that could be effectively treated with entrectinib or larotrectinib.

Answer: NTRK fusion mutation and possible treatment was included.

Finally, I do not agree with the statement on line 133 "The association of immunotherapy with targeted therapy is widely used to overcome resistance", since the only combination approved  is  atezolizumab + cobimetinib and vemurafenib which  resulted more effective than vem-cobi alone in the first analysis, but recently, at a longer follow-up was reported as not more effective on survival than combi + vemu alone.  Furthermore, the associations of anti PD1 and targeted therapies are less effective than ipi-nivo combinations and more toxic.

Answer: Thanks for the comment, the phrase was modified, “widely” was changed for “can be” to overcome resistance, but with no expressive results at a longer follow-up.

Round 2

Reviewer 1 Report

Comments and Suggestions for Authors

The manuscript has been improved and all concerns raised by the reviewer were addressed, therefore it can be published.

Comments on the Quality of English Language

 Moderate editing of English language required

Author Response

The manuscript was sent to a professional for English editing.

Reviewer 3 Report

Comments and Suggestions for Authors

Line 582: I think the statement "targeted therapy is frequently used in NRAS-mutated melanoma" is incorrect, all studies have been disappointing and MEK inhibitors were not registered with this indication.

I will suggest a bit more caution in the possible clinical role of KMT inhibitors in solid tumors. In melanoma but not only, even also in Merkel cell carcinoma clinical Phase II  and Phase III studies did not confirm the expected efficacy.

Comments on the Quality of English Language

there are some typing errors  to check

Author Response

The comment was excluded. And we were more cautious on affirmations about KMT.